# "This Clock Is Oop" Development of Translanguaging Strategies and Language Attitude in a Deaf Bimodal Bilingual Student

**Rhys McGovern**

New Leaf Language Independent Scholar, 1062 KS Amsterdam, The Netherlands; newleaflanguage@gmail.com

**Abstract:** Bilingual students do not always use language exactly the way teachers expect. This is especially true for deaf students who use a signed language and a spoken language. When students mix languages, they are often told they are making mistakes. Instead of forcing students to keep languages completely separate, teachers can let students use any and all of their language tools. When students can practice language with all of the languages they know, they feel better about their language skills. This article describes the language of one deaf bilingual student over one year. The author taught the student how to carefully examine and discuss his own language. The student then wrote about pictures and videos that he was interested in. At the end of the year, the student was comfortable using many types of language tools. He also felt powerful about his own language skills.

**Keywords:** translanguaging; bilingual; bimodal; deaf; language; metalinguistic; pedagogy; attitude; agency

## 1. Introduction

Bilingual students are not simply learners and users of two separate languages, but rather holders of a complex, flexible "unitary network of meaning" that incorporates multiple named languages (Garcia et al. 2021). The term 'translanguaging', which originally referred to inclusion of a minority language alongside a majority language in the classroom, has come to encompass a wide variety of concepts related to the behavior and pedagogy of multilingual language users and education (Swanwick 2017; Jaspers 2017). An important foundational concept of translanguaging research and pedagogy is that within an individual's multilingual system, there are no boundaries drawn between different named languages. However, strict social and cultural conventions of where one language starts and the other ends are frequently imposed on multilingual speakers. The term 'translanguaging' is used here to describe the bi- and multilingual practice of fluidly accessing and moving between and across named languages (e.g., American Sign Language, Spanish, English). Translanguaging is understood to occur across communication environments (Solorza et al. 2019) and across language modes (signed, written, or spoken). The title image of this paper, shown in Figure 1, is a representative example of translanguaging employing English and American Sign Language (ASL). The student wanted to express the concept of 'wrong/incorrect', but her linguistic repertoire did not include the English word 'wrong'. Drawing on her full linguistic repertoire and moving beyond the conventions of written English, she phonetically transcribed the mouthing 'oop' that commonly accompanies the conceptually equivalent ASL sign.

In monolingual environments, especially language education environments, translanguaging practices and strategies are frequently treated as errors and counted against a student's linguistic skill. 'Mixed' language expressions such as the title example may be misunderstood by teachers as lack of understanding, or intentional dismissal, of target language conventions. When teachers are not fluent in the named languages their students use, these misunderstandings may be more frequent. Far from being mistakes, the

translanguaging practices of bilingual learners can be understood as expressions of a robust semiotic repertoire comprising multiple named languages. For deaf bimodal learners, translanguaging practices across language modes are reasonable as the manifestation of natural cross-modal linguistic activation (Lillo-Martin et al. 2016; Villameriel et al. 2022). Further, when translanguaging practices embedded in the experience of bimodal bilingual learners are recognized, they can be leveraged to contribute to literacy development (Swanwick 2000; Podadera 2017).

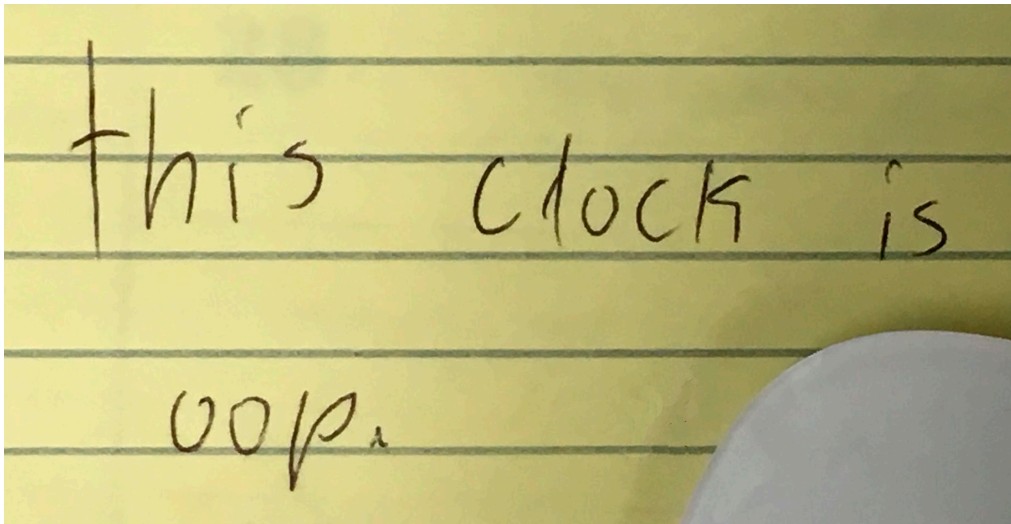

**Figure 1.** A piece of yellow lined paper with the handwritten sentence 'this clock is oop'.

This paper describes the process of the author's work with a bimodal bilingual deaf student enrolled in a subseparate bilingual high school mainstream program for deaf and hard of hearing students. Snapshots are presented from one academic year of weekly individual ASL/English language therapy and instruction sessions. A metalinguistic strengths-based pedagogical approach elicited the student's increased attention to, engagement with, and knowledge of his own translanguage system. Developments in his translanguaging strategy repertoire and level of independent strategy use are discussed, as well as developments in his literacy skills and conventional use of written English. In both the case study data and wider descriptions of methodology, attention is given to language attitude and self-image as a metric of linguistic agency. Instructional strategies included building and using metalinguistic vocabulary, modeling translanguaging practices, creating open space for the student to employ and explore his own translanguaging practices, and guided self-review and analysis of language production.

## 2. Translanguaging and Deaf Education

Deaf education in the United States and around the world has a long history of encouraging students to conform to linguistic expectations of the hearing world through discouragement of signed language and use of spoken language (see Baynton 1996; Burch 2002; Snodden 2020). While the cultural value of sign languages has been more widely recognized (Leigh et al. 2018), the fight for all deaf children to have access to full language and education is ongoing (Snodden 2020). Alongside the call for fully accessible schooling and language development for deaf children, translanguaging practices must be recognized as an integral part of the bimodal bilingual experience.

In an environment of reluctant or absent recognition of translanguaging strategies, language attitude and motivation can be affected and impact the outcomes of language learning (Lasagabaster 2012). When translanguaging behaviors are ignored or dismissed, students may internalize and follow the expectation of strict linguistic separation, thus limiting their demonstration of linguistic skill (Velasco and Garcia 2014; Solorza et al. 2019).

This necessitates suppression of natural language behaviors (see Lillo-Martin et al. 2014; Swanwick 2017) and can create an attitude of linguistic futility and deficit-based thinking (Velasco and Garcia 2014). Both these language attitudes are seen in the student case described here.

A difficulty of integrating translanguaging theory and pedagogy with deaf education is that bimodal translanguaging can look very different to unimodal translanguaging. In the title example, the student needed to convey a conceptually equivalent sign for an English word without having access to a conventional writing system for ASL. She was then faced with the question of how to represent a three-dimensional semiotic unit in two-dimensional graphemes, and creatively chose to use a portion of the sign—the movement of the mouth—that would be recognizable to a bimodal bilingual reader. This is translingual synonymizing: replacing a word in one named language with a conceptually equivalent or similar word in another named language. In bimodal bilingual production, expressing the conceptual equivalent can require more consideration than a simple word substitution. Translingual synonymizing was used frequently by the student described in this case study, and examples can be found in the included work samples. Other translanguaging strategies are also presented, such as self-talk or rehearsal in the nontarget language and application of syntactic conventions from one named language to production of another named language. Backtranslation, another strategy seen frequently throughout this study, can be understood as synonymizing on a larger scale, moving repeatedly between named languages to clarify and confirm production at the phrase, sentence, and paragraph level. While each of these strategies can easily result in messages that contain errors in conventional language use, they indicate strong and flexible translanguaging systems.

However, the concept of translanguaging, when applied to deaf education, must acknowledge the long struggle of minority language recognition and the pedagogical presence of signing systems such as Simultaneous Communication, also known as SimCom or Sign Supported Speech (De Meulder et al. 2019). SimCom, as discussed by Emmorey et al. (2008), is a system frequently used by educators of deaf students, which produces linguistic messages in both signed and spoken language. SimCom is a forced language mixing—both modes must be produced simultaneously—that typically produces messages within a spoken English syntactic frame when used by hearing educators (Emmorey et al. 2008).

Alongside other created signing systems, SimCom persists as a pedagogical approach in deaf education despite longstanding critiques of its learning outcomes and use (Hoffmeister 2000; Nover and Andrews 2002; Tevenal and Villanueva 2009; Swanwick 2017; Wang et al. 2017; Scott and Henner 2021). When the term 'translanguaging' is used to describe SimCom behaviors of teachers of deaf students, it "often hides power dynamics that exist between speech and sign, with sign language grammar and vocabulary placed at risk of vanishing if sign and speech are used together by hearing teachers who are not fluent in sign language" (O'Neill 2017). This is not equivalent to language production in bimodal bilingual children and adults, which naturally involves translanguaging in the form of cross-modal language mixing and which responsively uses ASL or English as the dominant syntactic frame (Emmorey et al. 2008; Lillo-Martin et al. 2014, 2016).

Examples of SimCom as an expressive translanguaging strategy can be found in the following case study. The reader is urged to understand this as one of many natural cross-modal strategies employed by the student, and not as a communication system routinely used in language instruction. As Swanwick (2017) states, it is crucial that translanguaging and SimCom not be conflated, from both pedagogical and language policy perspectives. For a review of evidence and arguments regarding SimCom within deaf education, the reader is referred to Scott and Henner (2021). Here, SimCom is recognized as one of many expressive translanguaging behaviors, not propounded as a pedagogical approach.

## 3. Author Positionality

The subject of this paper is connected to my personal experience as a bimodal bilingual, hard of hearing speech language pathologist and teacher. As a daily user of signed and

spoken language, discussion of shared experiences and the relationship to deaf cultural identity are frequently part of my work with deaf students. While deaf cultural identity and disabled identity are central to my lived experience, I recognize the linguistic privilege of being born hearing to a hearing family, and of losing hearing later in my life. I also acknowledge the limitation of my perspective as a white, educated person born and raised in the United States.

### 3.1. Methods: Study Model, Principles, and Setting

This case study offers a longitudinal description of bimodal language work with one student. Unlike studies that seek to identify trends in group data, a longitudinal case study allows for exploration of individual particularities and can describe the specifics of growth and responsive pedagogy over time. The student described in this paper experienced a variety of cognitive and linguistic challenges; from an educator's perspective, such a complex profile can be overwhelming, and can make it difficult to envision approaches suited to the individual's needs. While this paper is not intended to present a method or solution that will be applicable to all students, I hope each reader, especially those who work with students with complex profiles, will find concepts or strategies to support their work.

The CUNY-NYS Initiative on Emergent Bilinguals recommendation (Solorza et al. 2019) of a translanguaging classroom model includes deep individual assessment in each named language, design and delivery of how translanguaging can support the development and growth of each named language, a translanguaging space in which students are able to use their full linguistic repertoire, and a culminating project in one or both named languages. The principles of this model were applied at an individual instruction level. For the student in this case study, formal and informal assessments in ASL and English were conducted by the author prior to collection of work samples. A brief qualitative summary of the student's relative cognitive and linguistic strengths and challenges is provided as part of the case study introduction. Translanguaging design, delivery, and projects were then carried out by the author in microcosm during 60 min individual language therapy and instruction sessions throughout the year. The work samples and self-reports included in this case study are shared with permission of the student.

### 3.2. Methods: Culture Building

In order for students to bring all of their translanguaging practices to the room—and to the table—they must feel that all parts of their language repertoire are welcome, including strategies and habits that may have previously been identified as mistakes. "Bilingual students learn very early that their language practices are not welcomed in school, and that their translanguaging practices are not appreciated even in DLBE [dual language bilingual education] classrooms that are supposed to support them" (Solorza et al. 2019, p. 24). While this paper focuses on a case example of a single student, the process of culture building is first described as it includes all the deaf and hard of hearing students entering the language therapy room.

As research suggests that students' language attitudes are determined by their own experiences of language use (Tódor and Dégi 2017), and deaf students often have a long history of negative language experiences, the guiding aim of language therapy was to provide opportunities for affirming language experiences and develop linguistic agency. The language therapy classroom was established as a space in which all language behaviors and experiences, including those which exist outside the conventional bounds of named languages, were explicitly welcome. 'Crip linguistics' was explored in depth by Henner and Robinson (2021) and imbued all the work carried out in the language therapy classroom. This author's experiences of 'betweenity' were frequently pedagogically incorporated into conversation, accompanied by natural bimodal translanguaging behaviors.[1] In addition to teacher identity as pedagogy, three primary strategies were used to develop an acces-

sible environment that invited linguistic creativity and vulnerability from each student, regardless of the language modes or named languages present in their repertoire.

First, all named languages were recognized and made visible as part of the environment, allowing students to claim agency and see themselves reflected in the learning space. When students entered the room at the beginning of the school year, they were engaged in casual conversational interviews. They were invited to share about the language(s) they used and which were used around them at home, at school, and in social and community settings. These conversations included space for students to share their perceptions of their own communication efficacy in different settings, as well as examples of particularly challenging or successful communication memories. Each student was then invited to contribute to a 'community language board' by writing or drawing about their language system on index cards. They could make as few or as many as they desired; representations included pictures, drawings of ASL signs, and written English, Korean, Mandarin, Haitian Creole, Portuguese, Polish, and Spanish. This spread of cards remained on a bulletin board for several weeks, inviting additions and sparking conversations in and outside the classroom on language use, learning, and culture.

Second, students were continually offered opportunities and tools to employ their full linguistic repertoire, including with exercises that targeted production in one named language. Each student was introduced to and given supported practice time with a variety of tools, such as bilingual dictionaries, using a whiteboard for large-form multilingual brainstorming, and a computer with a camera to capture expression in both ASL and spoken language. For example, students who used ASL or signing systems were introduced to strategies of notating signed language to draft, remember, and share their ideas. These tools were available at any time, and students frequently accessed them outside of their scheduled language therapy sessions. When working with a task eliciting written English, a student might ask clarifying questions or check spelling of an English word in ASL. For a task eliciting signed storytelling, another student might draft their story using a combination of pictorial representations of signs and key words in English. As students grew more familiar and independent with their own language strengths and needs, they refined their choice of tools and required less support to use them successfully.

Finally, translanguaging practices were modeled and discussed as valuable, naturally occurring behaviors throughout work in the language therapy classroom. Further, translanguaging strategies were discussed in the wider context of multilingual people engaging with monolingual environments and expectations, and especially of deaf individuals navigating a hearing world. In everyday contexts requiring flexible communication behaviors, deaf and deafblind individuals make use of every mode and tool immediately available to them (Kusters 2020). This was demonstrated to students through frequent exploration of the varied communication strategies and tools they and the author used throughout and beyond the school day. By exploring examples of translanguaging in non-instructional interactions, students were encouraged to recognize translanguaging strategies as a powerful part of their ever-present language systems.

### 3.3. Methods: Individual Intervention

The culture-building work of the beginning of the year naturally extended into one-on-one work with each student during language therapy sessions, becoming highly individualized to each student's background knowledge and linguistic skill level. When translanguaging processes emerged in student work, they were consistently and explicitly named as linguistic strategy use. Multimodal resources and strategy use beyond signing, writing, and speaking—such as drawing, video, and movement—was modeled through instruction and encouraged in student practice. The case of one deaf bilingual-bimodal student is presented below to enable readers to envision individual translanguaging instruction, modeling, and guided practice.

Examining translanguaging behavior relies on a foundation of metalinguistics—one must have general knowledge of language forms, be aware of their differences, and use that

to construct abstract representations of language (Bialystok 2001). As Borman et al. state, in metalinguistic practice "language itself becomes an object of thought" (Borman et al. 1988, p. 325). Bilingualism alone does not unequivocally affect the development of metalinguistic skills associated with literacy (Besse et al. 2010), but metalinguistic knowledge development in deaf children has long been an area of educational concern (see Borman et al. 1988; Novogrodsky et al. 2014). Assessing and building a foundation of metalinguistic awareness and ability was at the core of translanguaging work described with this student.

*3.4. Case Study: Jared*

Jared was a deaf high school student from a hearing English-speaking family. His educational records indicate he received early access to hearing assistive technology and was exposed to sign-supported speech during preschool and early elementary grades. Jared's strongest receptive and expressive language avenue was ASL, but he expressed discomfort and embarrassment about signing in front of his hearing peers. He was heavily involved with school sports and his nonlinguistic social navigation skills were excellent, but he admitted to bluffing through most of his communication interactions in English with peers and teachers.

Like other bilinguals, Jared was attentive to the languages used by his communication partners (Gampe et al. 2019; Ortega and De Houwer 2019), and typically produced only spoken language with nonsigning peers and adults. However, in mixed deaf/hearing spaces and with individuals who signed, Jared's most frequent language behavior was producing a mix of signed and spoken language simultaneously. Throughout his SimCom productions, the bulk of Jared's message was alternately conveyed by his hands or through his voice depending on where his stronger conceptual vocabulary and syntactic skills fell in each moment. While this approach demonstrates extraordinarily rapid language choice making to engage his full linguistic repertoire, his messages were often difficult to understand in their entirety.

In the academic year before the work presented in this case study, Jared's expressive and receptive language and communicative problem-solving skills were assessed using informal and formal tools in ASL and English. The results of that evaluation describe that Jared was able to meet most of his daily communicative needs, with particular strengths in social communication with peers and adults around familiar topics. However, he struggled with the vocabulary and syntax needed to accurately express and understand abstract and complex concepts, such as those he encountered in the classroom or with unfamiliar social topics. Jared's narratives of his own and others' experiences were out of sequence, decontextualized, and difficult to understand. He also experienced challenges with self-advocacy, clarity of expression, and understanding others when communicating in unfamiliar situations or with unfamiliar listeners. Jared was most successful in both ASL and English when information was presented in reduced length and complexity, and when he was provided visual supports.

In addition to the language assessment results, neuropsychological testing results identified that Jared contended with severe challenges in working memory and temporal reasoning, abstract concepts, and learning new information. At the time of evaluation Jared did not consistently acknowledge his areas of difficulty or request communicative accommodations, though he frequently reported experiences of frustration, struggle, and shame at school related to his social and academic performance. Jared's linguistic, cognitive, emotional, and social profile was suggestive of lasting effects of early language deprivation (see Gulati 2018).

While it may be argued that such aspects of a student's profile present confounding factors that interfere in the analysis of bilingual presentation, the complexity of Jared's case is representative of many deaf students this author has encountered. Most deaf children are born to hearing families who cannot immediately provide them with fully accessible language (see Humphries et al. 2016; Hall et al. 2017), and wide differences exist in the quantity and quality of language access across early deaf education programs and

settings (LaSasso and Lollis 2003; Priestley et al. 2018). Limiting investigative discussion of ASL/English translanguaging only to a bilingual experience unaffected by reduced early language access omits the present daily reality of many deaf students and adults. This paper aims to contribute to the corpus of resources for teachers and researchers working with deaf and hard of hearing students, especially older students, with complex profiles.

### 3.5. Instructional Practices and Sampling

Language instruction was conducted primarily in ASL to draw on Jared's relative strengths and ensure optimal access to instruction. Metalinguistic approaches modeled and scaffolded by the instructor included making linguistic thinking visible through self-talk, close analysis of language form and structure, attending to and non-judgmentally drawing Jared's attention to his language output, and investigating the reasons for his language choices. Translanguaging strategies were embedded in this metalinguistic work through frequent translation and backtranslation of concepts and problem solving vocabulary and syntax obstacles as they arose.

Because Jared's profile included significant difficulties with sequencing and working memory, support for and strength-building with working memory was incorporated into language work. Stable visual supports were used heavily over the first six months of instruction by projecting high-interest images on a whiteboard as writing prompts. Components of the images were labeled, and the projected images and written vocabulary were left visible while Jared completed writing about the images. This approach invited Jared's engagement through student-led topic choice, capitalized on the accessibility of visual information, and provided targeted support for his working memory. As Jared developed his ability to manage information through strategies such as word banks, the writing prompts shifted to include other media, such as captioned video. All his writing was completed in a dedicated notebook, allowing him and the instructor to review and reference multiple drafts and previous work.

Each work sample presented in this case study was descriptively analyzed, revealing the sometimes small but crucial changes in Jared's linguistic repertoire and strategy use. The reader will notice that in some samples, Jared made more errors according to the conventions of written English than were addressed through the described instruction. While some aspects of his writing were addressed through vocabulary and syntax instruction in other sessions, some were not. With translanguaging and crip linguistics as pedagogical cornerstones, this language work prioritized meaningful (to the student) growth without haste or imposed flawless adherence to named language conventions. While all aspects of Jared's written work were discussed in the frame of communicative effectiveness, the choice of what components to address through instruction were guided by his communication desires and goals. This centers language as "a critical site of agency and an important form of care work." (Henner and Robinson 2021). Each work sample was descriptively analyzed, revealing the sometimes small but crucial changes in Jared's linguistic repertoire and strategy use.

Work samples were collected across nine months in one academic year, including prompt images/board work and samples of Jared's writing from his classroom notebook. Five writing samples are presented below; the first sample was collected in mid-October, the second in late November, the third in early January, the fourth in late March, and the fifth in late May. An additional non-writing example from late June is included to illustrate Jared's shift in language attitude and agency, and his growth in translanguaging strategy use. Language transcription from images uses (parentheses) when Jared added elements after his initial writing, and when he crossed elements out.

### 3.6. Sample 1: Early Work

Figures 2 and 3 show an early example of visual support and the written work Jared produced. The vocabulary on the board was written by the instructor during conversation about the picture in ASL, with the named strategies of thinking aloud in the comfortable

language of ASL and then representing ASL vocabulary in English. Written vocabulary was kept to single words and concretely linked with components of the picture through text location, size, and orientation.

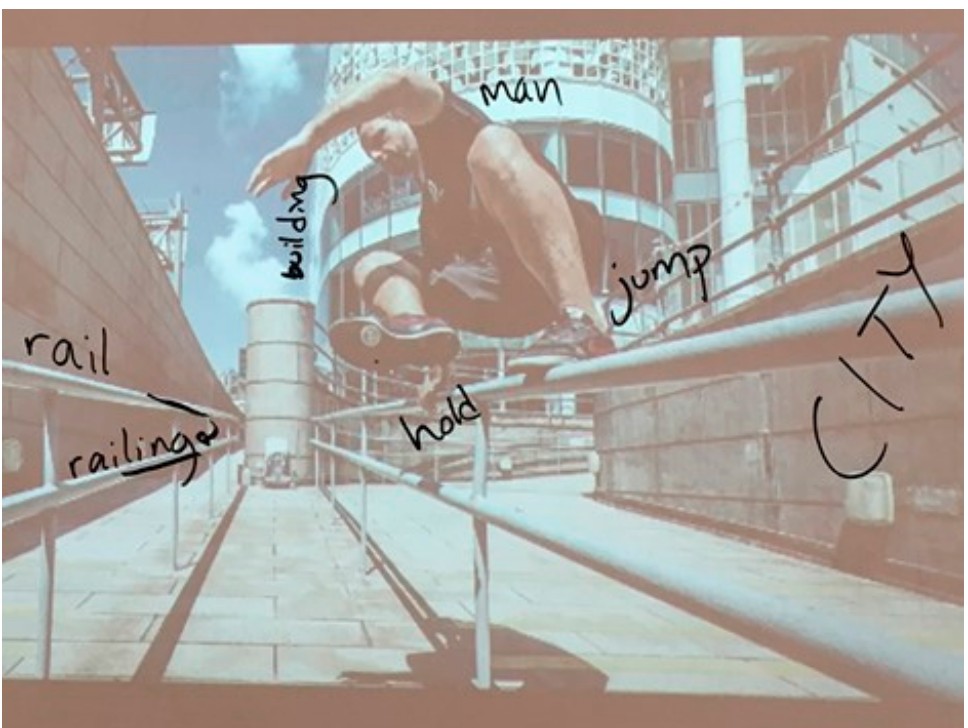

**Figure 2.** Projected image of a man jumping over a railing in an urban landscape. The English words 'rail', 'railing', 'building', 'hold', 'man', 'jump', and 'city' are written into the image on or immediately next to what they are labeling.

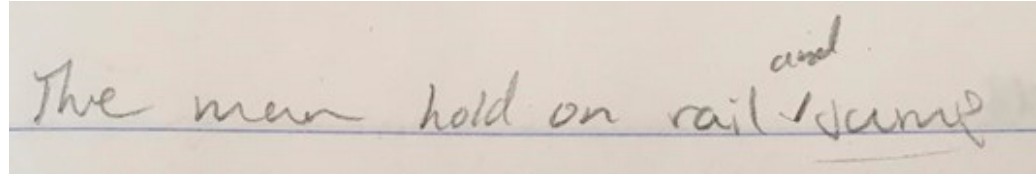

**Figure 3.** Jared's handwriting: The man hold on rail (and) jump.

Jared's written production consisted of a single sentence with at least three omitted obligatory morphemes. The instructor then initiated discussion of his writing in ASL with the question 'Is your writing in English showing everything you would express in a similar sentence in ASL?' Jared identified that he knew there were some words missing, but he was not sure what they were. The instructor then drew his attention to how his body position and sign rhythm showed 'hold' and 'jump' as two separate actions when describing the image in ASL, and Jared reexamined his sentence and added the word 'and' between the two English verbs. Concepts of verb clauses and conjunction use to link and separate clauses were topics of discussion for several following sessions.

*3.7. Sample 2: Increasing Sentence Length and Engagement*

In the second sample, the instructor continued to write vocabulary directly in the image, with text location as a consistent support (Figure 4). Text orientation and size are no longer used as supportive factors for vocabulary mapping. The vocabulary is also no longer only single words, but includes several multiword phrases. Jared was more engaged in the process, actively suggesting vocabulary labels through a mix of ASL

signs, spoken words, and fingerspelling. His increased engagement also appears through longer spontaneous writing (Figure 5) and direct use of terms from his personal linguistic repertoire (freerunning, parkour) that were not obligated through the image itself.

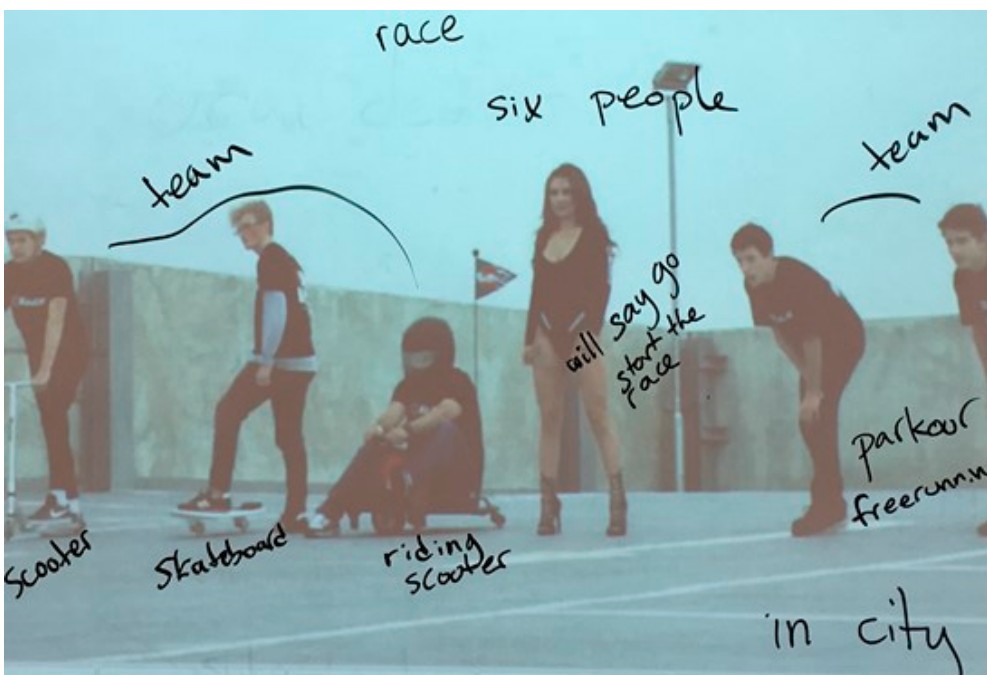

**Figure 4.** Projected image of six people standing lined up in the empty top floor of a parking garage, in race-start positions. The vocabulary scooter, skateboard, riding scooter, parkour, and freerunning are written as labels next to each of the five races, and the phrases 'will say go' and 'start the race' are written next to the woman in the middle of the image. At the top of the image are written 'race' and 'six people', and an arced line is drawn over the three people riding vehicles and the two people getting ready to run. The word 'team' is written over each arced line and the words 'in city' are written near the bottom of the image.

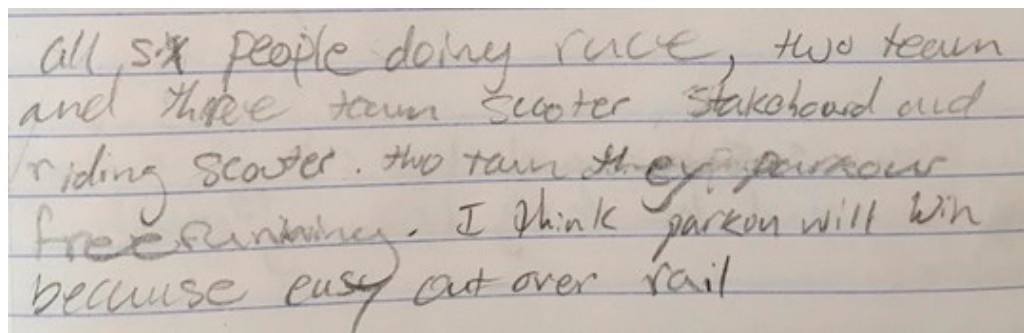

**Figure 5.** Jared's handwriting: all six people doing race, two team and three team scooter stakeboard and riding scooter. two team they parkour freerunning. I think parkou will win because easy out over rail.

As Jared wrote, he referenced the vocabulary on the whiteboard and rehearsed phrases in ASL using self-talk. He independently incorporated attention to his ASL phrasing, and used 'and' twice to connect separate concepts in his writing. When he finished writing, backtranslation (reading his writing one sentence at a time and producing conceptually equivalent ASL) was modeled by the instructor as a strategy for checking the message he had produced. One of the free morpheme types omitted in Jared's previous writing (conjugations of the verb 'to be') continues to be omitted here; state verbs that are lexically omitted in ASL but obligatory in English were the focus of several following sessions.

*3.8. Sample 3: Developing Agency and Self-Regulation in Writing*

In Figures 6 and 7, Jared's increased agency is evident: he took initiative during ASL discussion of the images to start mapping English vocabulary himself. During this language session, two images with similar content and increasing complexity were presented as prompts, and Jared produced writing for each one. In Figure 8, Jared's writing shows independent use of correct conjugations of the verb 'to be' twice. For the first image prompt, backtranslation was modeled again, and during that process the use of spatial anchoring in ASL was discussed as a conceptually equivalent option for prepositions in English: signing the location/object then signing an agent classifier to show "in the woods", "on the powder". After this discussion, Jared offered that his English writing was missing preposition concepts, and added the words 'in' and 'on'.

During discussion of and writing about Figure 7, Jared used self-talk to rehearse his sentences in ASL before writing in English, and in response to the question 'how much of your vocabulary [on the whiteboard] did you use in your writing?' he checked off the vocabulary terms for each image. He also circled several that he had used together in a sentence. Jared spontaneously produced backtranslation for his finished writing after working with the second image, and noted with satisfaction that he had included prepositions.

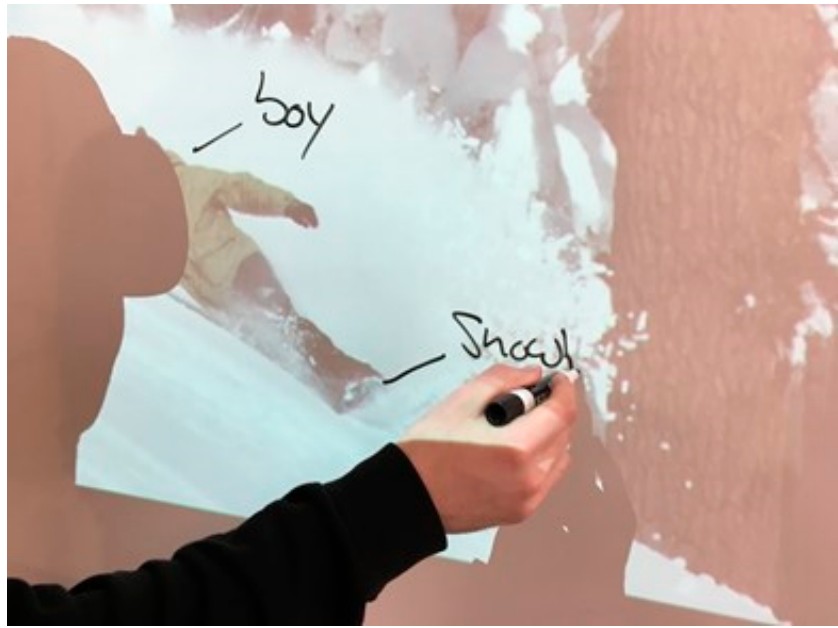

**Figure 6.** Projected image of a man snowboarding down a wooded mountainside, with the word 'boy' written with a line leading to the man. Jared's hand is visible as he writes the word 'snowboard' with a line leading to the snowboard.

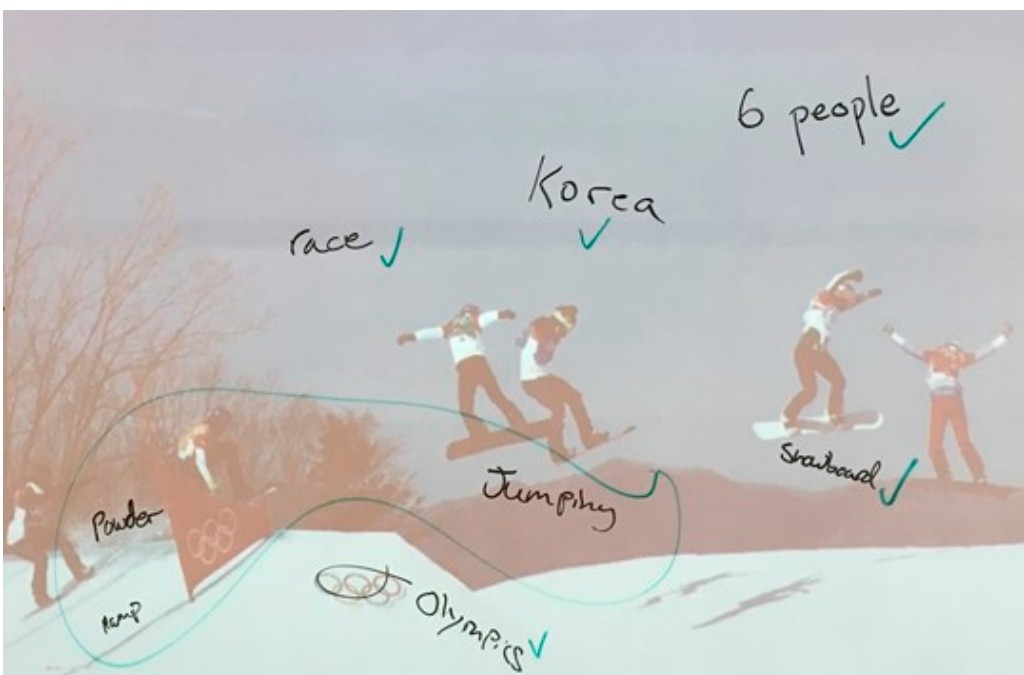

**Figure 7.** Projected image of six snowboarders on and in the air after a jump with the Olympic symbol on the side. The vocabulary 'race', 'Korea', and '6 people' are written above the people in the image, and the words 'powder', 'ramp', 'jumping', 'snowboard', and 'Olympics' are written into the image on or immediately next to what they are labeling. There are green checkmarks next to all the written words except 'powder', 'ramp', and 'jumping', which are grouped by a green loop drawn around them. All words except for the phrase '6 people' are in Jared's handwriting.

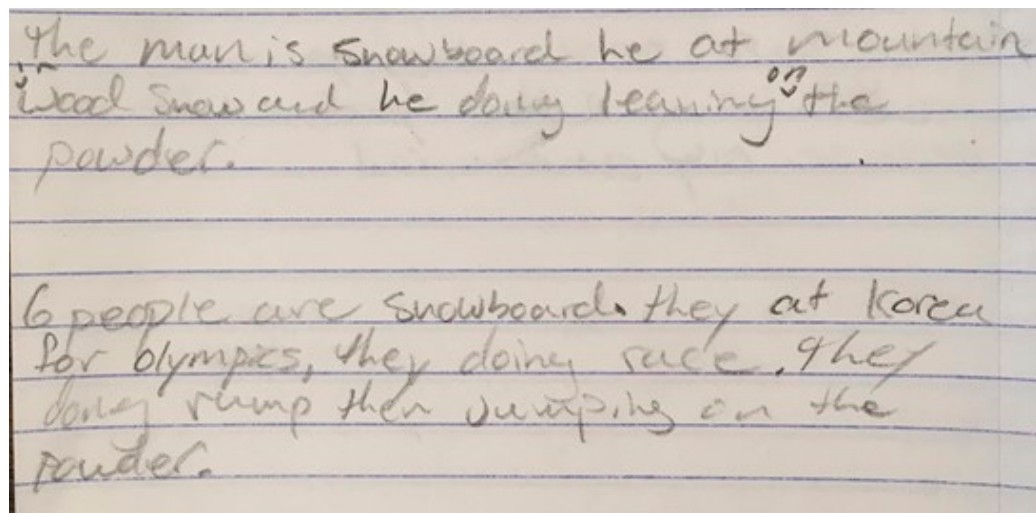

**Figure 8.** Jared's handwriting: The man is snowboard he at mountain (in) wood snow and he doing leaning (on) the powder. people are snowboard. they at Korea for olympics, they doing race. They doing ramp then jumping on the powder.

### 3.9. Sample 4: Independent Strategy Use

In Figure 9, Jared transitions to claiming full agency over the vocabulary transcription process: all the writing on the whiteboard is his own. He independently employed a translanguaging strategy that had been previously prompted: while mapping vocabulary onto the image, Jared paused, signed REFEREE, and asked for the equivalent English word.

After the instructor had fingerspelled it, Jared wrote it on the image and asked for a spoken model as well.

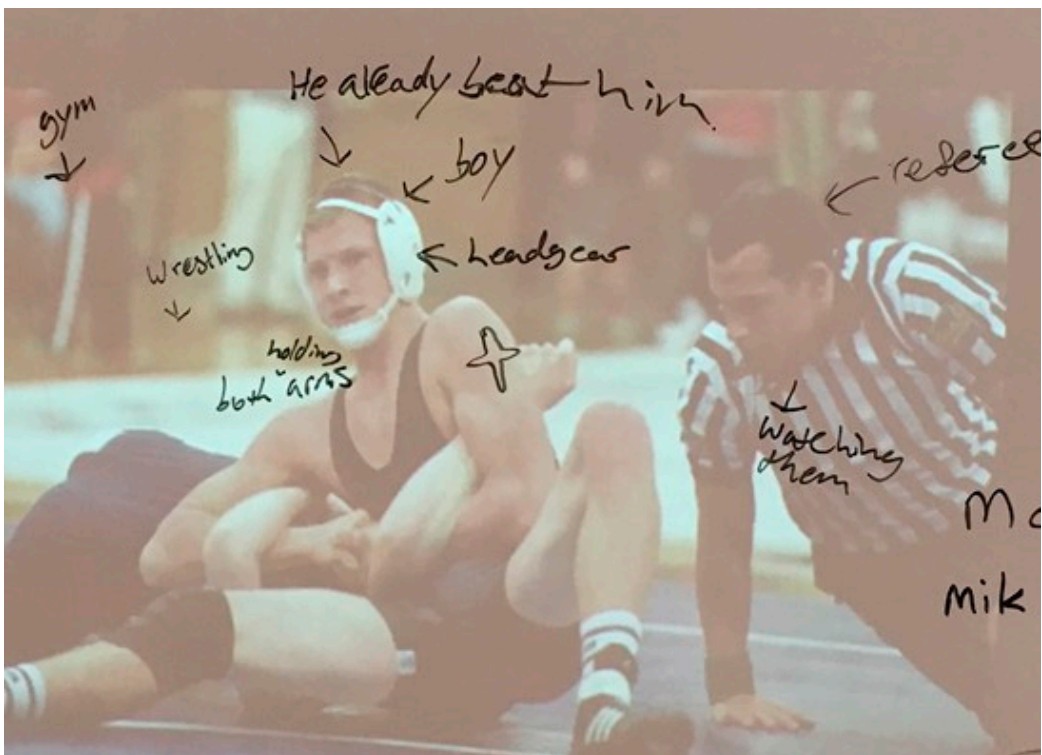

**Figure 9.** Projected image of two young men wrestling in singlets and headgear, with a referee in a black and white striped shirt crouching and watching carefully on the wrestling mat beside them. The words 'gym' and 'wrestling' are written to one side of the image, with arrows pointing down into open space behind the people at the center of the image. The vocabulary 'boy', 'headgear', 'holding both arms', 'referee', and 'watching them' is written into the image, with arrows pointing to what they are labeling. At the top of the image is the sentence 'He already beat him.' with an arrow pointing down to the wrestler holding the other in a lock on the mat.

In Figure 10, his written samples show an emerging process of drafting his writing toward a finished product. He consistently uses prepositions, conjunctions, and correct conjugations of the verb 'to be' in his draft paragraphs and his final product. The verb tense forms in Jared's writing are also becoming more sophisticated and varied, whereas before he had used almost exclusively present tense forms. The metalinguistic label 'word bank' was introduced at this point and used to describe the whiteboard vocabulary mapping, then explained as a strategy that could be removed from the context of picture support. Jared frequently created word banks for his writing after this session, using a facing page in his notebook to have the vocabulary close at hand as he wrote.

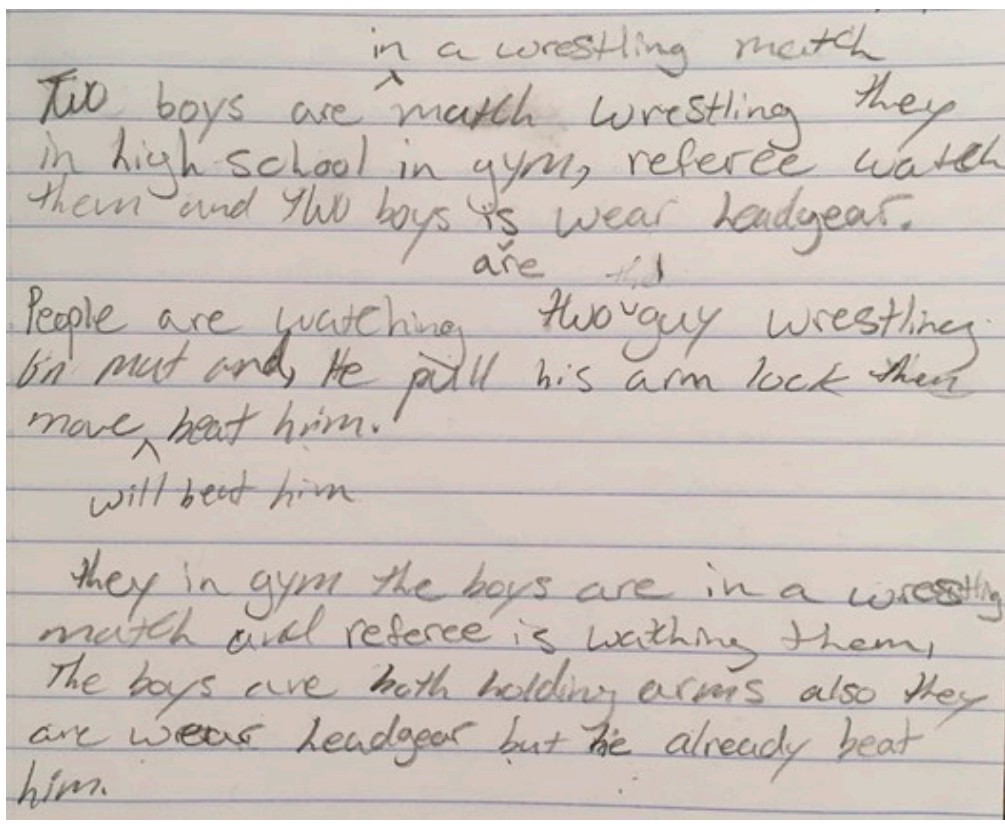

**Figure 10.** Jared's handwriting: Two boys are match (in a wrestling match) in high school in gym, referee watch them and two boys are wear headgear. People are watching two guy wrestling on mat and, he pull his arm then move beat him (will beat him). They in gym the boys are in a wrestling match and referee is watching them. The boys are both holding arms also they are wear headgear but he already beat him.

### 3.10. Sample 5: Flexible Strategies and Multiple Drafts

With the shift from working on the whiteboard to creating word banks in his writing notebook, Jared was supported in developing a more intentional drafting process; in Figure 11, Jared brainstormed and drafted sentences before attempting a paragraph. The writing prompts also increased in complexity, from images to video clips—in this case a pair of mechanics working on a damaged vehicle. Jared and the instructor frequently paused the video to discuss what was happening in ASL and add English vocabulary to Jared's word bank. In two instances, Jared used SimCom to produce phrases he was familiar with in English: 'started running' and 'roll cage'. For each phrase, the instructor responded with a direct imitation of Jared's signed phrase (START RUN, ROLL BOX) and asked 'Does that sign match the concept of the English phrase?' Jared was then able to revise his sign choices and build a conceptually accurate linguistic representation in both his named languages. When the clip was finished, Jared drafted sentences and then used his sentence drafts to construct a paragraph (Figure 12). In Figure 11, two drafts are noticeable: the first two lines are composed primarily of vocabulary terms loosely arranged into a narrative of the video clip. In the second section, Jared has refined his vocabulary and added obligatory free and bound morphemes (-ing, 'engine', 'they'). He also followed more conventional syntactic structure, including choosing the word 'replace' for his initial English transcription of the ASL phrase "doing are fix". Figure 12 also shows Jared's metalinguistic awareness that his crossed-out writing does not meet 'final draft' conventions; he rewrote the edited sentences cleanly and with full punctuation.

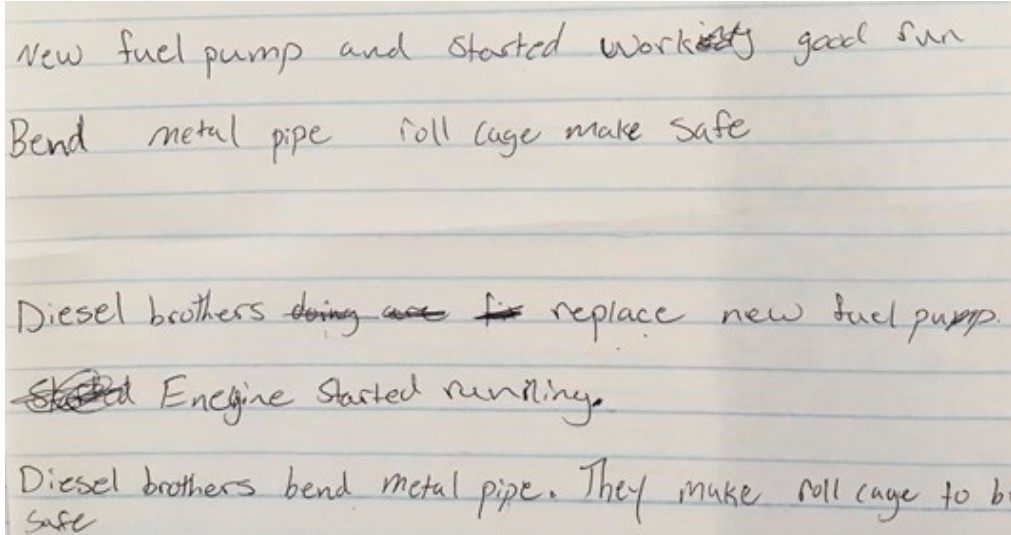

**Figure 11.** Jared's handwriting. The first two lines grouped: New fuel pump and started working good run/Bend metal pipe roll cage make safe. Neither line includes punctuation. The last three lines are also grouped, with periods ending the first two sentences: Diesel brothers replace new fuel pump. Enegine started running. Diesel brothers bend metal pipe. They make roll cage to be safe.

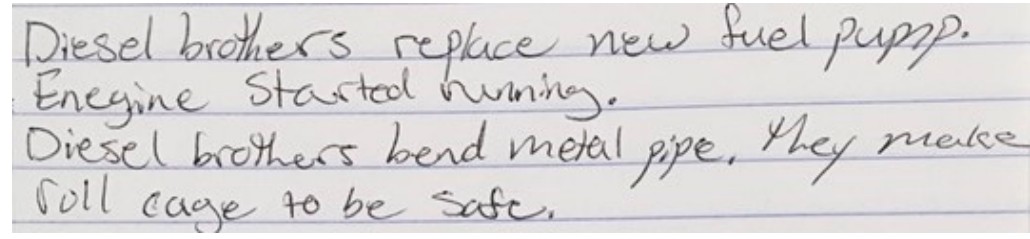

**Figure 12.** Jared's handwriting: Diesel brothers replace new fuel pump. Enegine started running. Diesel brothers bend metal pipe. They make roll cage to be safe.

*3.11. Sense of Agency and Social-Emotional Aspects of Language Practice*

Over the time of the documented language sessions, the effect on Jared's approach to writing and perception of his own language agency and capacity was enormous. At the time of the first work sample presented here, Jared frequently balked or outright refused to attempt writing or spelling words he did not know. He always closed the door of the classroom and pulled the door shade down to ensure no peers or adults saw him working, and showed minimal frustration tolerance for errors or uncertainty. He used harsh and punitive language toward and about his own language efforts, often describing his English (and sometimes his ASL) as 'bad' or 'wrong', and ended challenging language tasks nearly immediately with the futilistic response "I can't" or "I don't know."

In contrast, Figure 13 depicts a moment of translanguaging from close to the end of the year of documented work. When discussing this image and an upcoming plane trip he would take, Jared was attempting to communicate the concept of turbulence. Writing on the board, he attempted spelling of the word, which he had only heard spoken before but never seen printed, and his uncertainty is apparent in his writing. When his writing was unsuccessful at conveying his message, he attempted to speak the word several times, and when that was still unsuccessful he drew a circle around the cloud and added visual support of a wobbling line, at the same time signing a plane bumping through the air. This strategy successfully delivered his concept, and the instructor responded by imitating Jared's bumping plane sign, offered the spoken word "turbulence" to confirm the meaning, and fingerspelled the word before writing it in English on the board under Jared's first attempt.

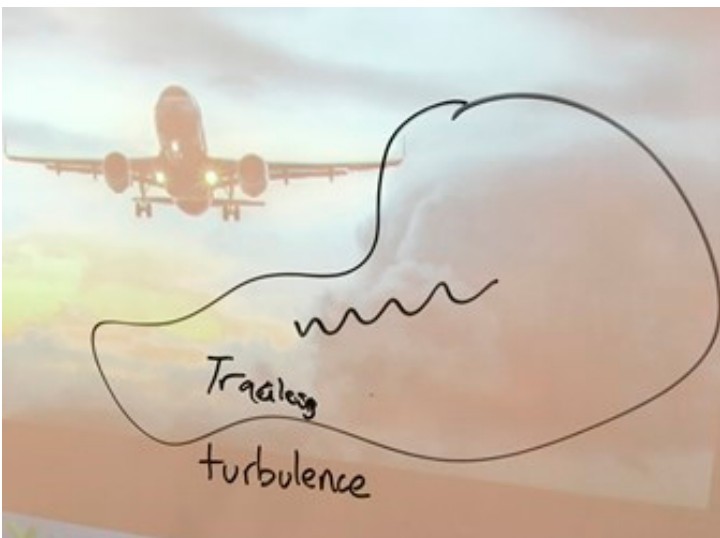

**Figure 13.** Projected image of an airplane surrounded by clouds. A large loop is drawn around one cloud, with a wavy line through the cloud and the written word attempt 'traaless'. At the bottom of the image is the written word 'turbulence'.

In that session, Jared persevered through several vulnerable moments of linguistic difficulty to engage his full bimodal translinguistic strategy repertoire, including spelling and speaking an uncertain multisyllabic word. When he successfully conveyed his concept and received the feedback of the correctly spelled English word, Jared expressed excitement and a growth-oriented attitude. He immediately practiced speaking the word several times, asking the instructor for support with syllabification and refining his pronunciation. He showed confidence in his internal concept representation, a sense of personal power and communicative effectiveness, and none of the shame that used to accompany his uncertain attempts.

## 4. Data Synthesis and Discussion

Between the first and last writing samples presented here, Jared developed more complexity, accuracy, and volume in his written work. While there are still obligatory free morphemes missing in his final writing, the complexity and comprehensibility of his messages increased, and he became more skilled, confident, and independent in his identification and analysis of language conventions. Here, a benefit of longitudinal data collection when working with bimodal bilingual students within a translanguage framework becomes clear: without knowledge of the preceding strategy development and skill building, Jared's final work sample could be seen solely as English writing containing multiple errors.

Jared's development of translanguaging and metalinguistic strategies deserves attention distinct from his written improvements. While Jared was clearly utilizing translanguaging strategies before this work began, he did not have a metalinguistic vocabulary with which to approach them. His characterization of his own language productions as incorrect demonstrated a noticeable sense of futility in his language attitude. However, when he came to trust the presence of accommodations and receive the explicit message that all parts of his language system were valuable, Jared developed facility with a variety of translanguaging strategies that supported his learning and conferred a sense of agency and linguistic power. He also incorporated external metalinguistic structures such as word banks, engaged in increasingly complex discussions of his own language productions, and grew adept at monitoring his own signing and writing for moments when he might ask for support.

Jaspers (2017) asks "How do you make pupils learn a collectively valued register without implying that their individual linguistic skills are less important?" I argue that

providing a regular space where accommodations are openly available and accessed, translanguaging is a welcomed norm, and communication practices are collaboratively examined and contextualized offers students the opportunity to safely engage and discover power in their full linguistic repertoire. However, when students are not introduced to their own language skills as valuable tools in which it is worth investing time and practice, they can easily internalize cultural monolingual standards and develop a sense of shame, frustration, and impotence about translanguaging.

## 5. Conclusions

For Jared, being introduced to a translanguaging framework and strategies in bimodal language work contributed to improvements in his conventional written English and positive shifts in his overall approach to language. While the quantitative shift in syntactic accuracy described here might not be captured in many classroom assessments, the qualitative effects of adopting a translanguaging framework were significant. At the time of the first work samples presented above, Jared was a hesitant, disinterested writer who was easily daunted by the challenges of putting his thoughts into written word. Over the year of work documented here, Jared moved from producing writing that he was not able to independently monitor and redraft, and that did not communicate his full meaning, to efficacious paragraphs that he independently examined and edited. He adopted an attitude of flexibility and growth that allowed him to meet challenges and feedback with resilience, and he developed language strategies to support his system in multifaceted ways (such as using word banks to monitor his concept inclusion and to support his working memory). Jared was more engaged, demonstrated an increased level of independence and sense of agency, and developed a level of resilience that had previously been inaccessible to him. In short, translanguaging practice brought him closer to being able to "deal confidently and competently with any real-consequences writing task" (Fox 1993, p. 109). It is the intention of this author not to add to what Jaspers (2017) describes as a demand for sole dedication to translanguaging practice, but to illuminate potential academic and social-emotional effects of an approach that does not penalize translanguaging behavior.

In warning against applying a proscriptive translanguaging framework to deaf education, De Meulder et al. (2019) posit that enacting and studying translanguaging practices with deaf students "requires attention to ideologies applied to these sensory orientations." Throughout this pedagogical incorporation of translanguaging practice, discussions engaging with those ideologies were frequent and rich. Translanguaging patterns in writing, signing, and speaking became a lens through which to begin exploring social and linguistic systems of power. Perennial topics included dominant cultural narratives and attitudes toward bimodal bilingual language, as well as the nature and scope of consequences of intentional and unintentional linguistic choices, especially those that go against monolingual norms. Confronting the idea of translanguaging practices as 'errors' necessitated confronting the monolingual ideology of English as more 'valuable' or 'useful' than ASL. Further, fostering a sense of linguistic agency and power allowed for the development of sensori-cultural resilience with which to resist such ideology.

All language behavior, including that which exists outside the conventional bounds of named languages, can exist as powerful communication. When students receive the message that their language use is incorrect, inappropriate, or inadequate, it can deeply impact their engagement, motivation, and self-image as language users. Furthermore, there is a missed opportunity for deep linguistic development across their named languages and metalinguistic engagement with their own language system in wider cultural and linguistic contexts. Approaching a student's language production without judgment or denigration, even when it 'breaks the rules' of their languages, can support increased self-confidence and agency in language use. Beyond bringing ASL education into equivalent importance to English education, language instruction within deaf education at any level must be designed for and include affirmation of deaf students in their full multilingual identities.

**Funding:** This research received no external funding. The APC was funded by the University of Connecticut Neag School of Education.

**Institutional Review Board Statement:** Not applicable.

**Informed Consent Statement:** Informed consent for use and publication of data was obtained from all subjects involved in this study.

**Data Availability Statement:** Not applicable.

**Conflicts of Interest:** The author declares no conflict of interest.

## Note

1   For more on deaf experiences of 'betweenity', see Brueggemann (2009). For more on identity as pedagogy in multilingual classrooms, see Morgan (2004).

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
