# Peer review of "“This Clock Is Oop” Development of Translanguaging Strategies and Language Attitude in a Deaf Bimodal Bilingual Student"

_languages, doi:10.3390/languages8010034_

Round 1

Reviewer 1 Report

Review

”This Clock is oop” Development of Translanguaging Strategies and Language Attitude in a Deaf Bilingual Bimodal Student

Overall review

The article presents a descriptive case study of instruction for a deaf bimodal bilingual student in a high school within the framework of translanguaging. The study describes how English is taught to a deaf student using translanguaging practices; i.e., ASL and English (spoken/written) are used
somewhat simultaneously. In addition, pictures are used as a scaffolding tool for further interaction between the instructor and the student. The article provides insights into how to work with ASL and English to develop the student's writing skills and provides an example of a development trajectory.
Although the limited generalization of the results provided here, the study is an excellent example of the strength of using a descriptive case study method as it provides new and unique knowledge in teaching English to deaf students using ASL. The samples were exciting and valuable. Also, this will make an interesting contribution to the area of translanguaging in general. The article is fascinating and valuable, but I have some revision suggestions for the author to consider.

Major comments

Method section. There is a need for a more expanded method section explaining the strengths of case studies, the setting in how data was collected, who the instructors were, etc. I am not convinced that
the sections can describe assessment results more broadly. The author needs to be more transparent here. What assessments were you using, and how did you conclude that Jared is language deprived? How long time was it between the samples, by the way?

Background. I think it would help the readers if you could explain some of the core issues of "translanguaging practices and strategies." Some examples would be helpful.

Minor comments

Title. I believe the first sentence, "This clock is oop" may be too hard to grasp for many readers. Especially for those who are unfamiliar with sign language research. It is excellent but requires reading the article before understanding the title. In fact, I first wondered if it was a typo error. I suggest you consider a different, more transparent example, or at least mark out "oop" with /oop/ or something similar. And "Bilingual Bimodal" – the literature often uses "bimodal bilingual" – is it a conscious choice?
86-103. About SimCom. You probably want to refer to Scott and Henner (2020) too.
111-113. I find these lines just confusing. I read it as you are supporting SimCom as part of translanguaging. I assume you don't. Consider deleting or rephrasing/expanding. However, there are alternative labels for using "SimCom" as part of translanguaging practice, e.g., "intramodal translanguaging" in the literature.
174-186. The label "telegraphic language" is widely criticized. However, it is excellent to lift it in this context. At the same time, I cannot avoid thinking of the risk of reusing this old label. Therefore, I would suggest linking it to available research on second language structure phenomena as we find that
kind of "telegraphic" constructions in these learners.
209-212. Where is the qualitative summary of the assessment results? Do you mean in the section beginning at 213? Please be clear. In addition, I feel this paragraph is a bit unnuanced in terms of language deprivation description. Have you considered bringing elements from the discussions of
"initial systems" (Chen Pichler & Koulidobrova) and "crip linguistics" (Henner & Robinson) to this discussion?
228. "Like other bilinguals" – what do you mean? It was a bit unclear how Jared is like other bilinguals.
236-237. Understand by whom? Please describe the setting and the instructor(s) a little more in the method section. (See above.)
263-264. I became curious about how the instructor handled the "three omitted morphemes." It seems that they chose not to cover all the omissions here. Please explain a bit further. Maybe as a conclusion of the Sample 1 section.
280. It helps the reader if you add that text location is still a supportive factor.
Figure 5. In the figure, parkour is spelled as "parkour", same with skateboard ("stakeboard"). The figure text doesn't correspond with the text in the figure. You need to be clear here if the figure text is a transcription or not.
438. You mention the importance of longitudinal data collection. The method section needs to include a more detailed description of the longitudinal data collection. (See above.)

Reviewer 2 Report

This is a clearly-written description of a novel case study approach into translanguaging over a one-year period. Examples of how translanguaging were cultivated are discussed throughout. This is significant because, which this is a case study, it is a case study of a language deprived child likely with a similar linguistic repertoire to many other language deprived students. Disseminating this may help other practitioners understand novel ways to work with their students.

I do not have significant issues with the content as it is, but I do have two recommendations.

1. It would be helpful to have a positionality statement of some kind; who is the author and what is their relationship to Jared. For example, the fact that data were collected during "ASL / English language therapy and instruction sessions" makes me suspect that the author is an SLP, but this may not be the case. Further, it would be helpful to know the author's hearing status and the languages in their own linguistic repertoire , as this helps to contextualize how they worked with Jared. Hochgesang (2022; https://youtu.be/LqJJsNd-Ib4) described why discussing positionality in research is important.

2. Because data were collected longitudinally, it would be helpful to have approximate dates of when each example was collected so readers can get a sense of how much time passed between each example. We know data were collected during a school year, so the first example is likely around August and the last around May or June, but for the interim examples, if this is possible to add, I think it would be helpful.

Reviewer 3 Report

Thank you for allowing me the opportunity to review this article. I apologize for it being late. As with every review, I do not attend to grammar and spelling. My review also follows a P:L format where P is the page number and L is the line number.

P1: Abstract – A plain text version accompanied by a one page plain language summary would be nice if possible.

P1:27 – Perhaps more information that the idea of translanguaging is not necessarily moving between languages, but that there is no distinction between languages.

P1:38 – Is it creative tool use or again rather that the languages aren’t distinct

P2:48.- Are the pronouns used here the pronouns that the student themselves selected or the assumed gendered pronouns of the participant?

P3:72 – I’m not sure if bi-bi is more widely adopted now than it was before. Recall that post 1989 was the bilingual experiments of the 90s and the adoption of STAR schools that widefly failed. I think you can look at Nover’s reports from that time for more details.

P3:75 – You may also want to cite Scott & Henner 2020 here for the specific critiques.

P3:105 – V good. Very important to recognize simcom as a viable translanguaging strategy, but not as a pedagogical method. The method’s goal is to provide instruction in the dominant language with sign as a support – the translanguaging is more about how the languages overlap in the person’s repertoire.

P3:119 – Yessssssss

P3-140 – Whenever one says sparking conversation, I wonder what these conversations were

P3:147 – I’m not very good with pedagogy, but these strategies still seem like code switching rather than true translanguaging. Does one must write in English? Which English? What would proper translanguaging look like in writing for ASL signers? These questions I wonder

P4:173 – Vygotsky seems thrown in here without context. I get that it’s related to scaffolding, but I’m not sure that Vygotsky would be the best citation within a translanguaging context

P4:174 – I dislike the word use of ungrammatical here. It clashes with the otherwise positive paper on translanguaging

P4:176 – The kids don’t have rudimentary syntax though. They have a syntax system which has served them well, but does not align with the community expecations of what their syntax system should look like because they haven’t been given access to a community of language learners.

P4:193 – I think novogrodsky would agree that it’s not that the kids are delayed or gapped, but they really haven’t been given opportunities to think holistically about their languages.

P4:200 – I prefer Hall for the citation over Gulati. But nevertheless, this is an observation that hasn’t been merited by assessment. I think the writers should make that clear.

P5: 215 – Sign supported speech used here instead of simcom. Do the authors intend a difference?

P5-221 – I am not a fan of the language of deficit and deviance here with regards to assumed language deprivation syndrome. No assessments for the syndrome exists as far as I know.

P5-248.- Working memory deficits determined via what assessment? Corsi blocks? Does it mater?

P6 – I like the figure descriptions. It makes the figures accessible to blind and visually impaired readers. Commendable.

P8 – 313 – May be helpful for novice readers to define back translation, spatial anchoring, etc.

P12 – 401 – I know this isn’t the point of the paper, but is paragraph for me is critical as it examines the effects of negative language perception and commentary from teachers on one’s self esteem and self conception

P12 – 414 – This is a very nice example of layed languages – translanguaging, and successfully using the semiotic repertoire. I want this to be emphasized.

 Overall I felt this was an excellent paper with some attention needed to the language of deficit and deviance. Some engagement with the citations would be nice. 
